# Characterizing the Genomic Profile in High-Grade Gliomas: From Tumor Core to Peritumoral Brain Zone, Passing through Glioma-Derived Tumorspheres

**DOI:** 10.3390/biology10111157

**Published:** 2021-11-09

**Authors:** Martina Giambra, Eleonora Messuti, Andrea Di Cristofori, Clarissa Cavandoli, Raffaele Bruno, Raffaella Buonanno, Matilde Marzorati, Melissa Zambuto, Virginia Rodriguez-Menendez, Serena Redaelli, Carlo Giussani, Angela Bentivegna

**Affiliations:** 1School of Medicine and Surgery, University of Milano-Bicocca, 20900 Monza, Italy; m.giambra1@campus.unimib.it (M.G.); e.messuti@campus.unimib.it (E.M.); r.bruno13@campus.unimib.it (R.B.); r.buonanno2@campus.unimib.it (R.B.); m.marzorati@campus.unimib.it (M.M.); m.zambuto@campus.unimib.it (M.Z.); serena.redaelli@unimib.it (S.R.); 2Neurosurgery Unit, Department of Neuroscience, S. Gerardo Hospital, 20900 Monza, Italy; andrea.dicristofori@gmail.com (A.D.C.); clarissa.cavandoli@gmail.com (C.C.); 3NeuroMI, Milan Center of Neuroscience, Department of Neurology and Neuroscience, University of Milano-Bicocca, San Gerardo Hospital, Via Pergolesi, 20900 Monza, Italy; virginia.rodriguez1@unimib.it; 4Experimental Neurology Unit, School of Medicine and Surgery, University of Milano-Bicocca, 20900 Monza, Italy

**Keywords:** glioblastoma, GBM, glioma stem cell, peritumoral brain zone, genomic profile, array-CGH, copy number alterations

## Abstract

**Simple Summary:**

The genomic landscape of the stem cell compartment and peritumoral brain zone of glioblastoma is still incomplete. The key role of the stem component in tumor maintenance and progression, as well as in drug-resistance spreading, has already been demonstrated. In recent years, the importance of the marginal area of the neoplasm has been considered since this is where the tumor recurrences appear in up to 90% of cases. In this study, we carried out a 360° genomic profile analysis of the different glioblastoma components in order to understand how they are characterized and how they work. Studying their genomic constitution is the starting point necessary to finally develop target treatments associable with the standard ones, as a new hope for glioblastoma patients.

**Abstract:**

Glioblastoma is an extremely heterogeneous disease. Treatment failure and tumor recurrence primarily reflect the presence in the tumor core (TC) of the glioma stem cells (GSCs), and secondly the contribution, still to be defined, of the peritumoral brain zone (PBZ). Using the array-CGH platform, we deepened the genomic knowledge about the different components of GBM and we identified new specific biomarkers useful for new therapies. We firstly investigated the genomic profile of 20 TCs of GBM; then, for 14 cases and 7 cases, respectively, we compared these genomic profiles with those of the related GSC cultures and PBZ biopsies. The analysis on 20 TCs confirmed the intertumoral heterogeneity and a high percentage of copy number alterations (CNAs) in GBM canonical pathways. Comparing the genomic profiles of 14 TC-GSC pairs, we evidenced a robust similarity among the two samples of each patient. The shared imbalanced genes are related to the development and progression of cancer and in metabolic pathways, as shown by bioinformatic analysis using DAVID. Finally, the comparison between 7 TC-PBZ pairs leads to the identification of PBZ-unique alterations that require further investigation.

## 1. Introduction

Glioblastoma (GBM) is the most common and malignant primary brain tumor, characterized by rapid progression, invasion, high genomic instability, intense angiogenesis and resistance to therapies [1]. Despite aggressive standard treatments, the prognosis remains extremely poor with a mean survival of 20.9 months [2]. The failure of current therapies is mainly due to the striking inter- and intratumoral heterogeneity of the disease [3,4,5], supported by the presence, within the tumor mass, of cells with stem-like properties, called glioma stem cells (GSCs) [6,7]. To complicate this picture, there is also the presence, even if there is still a role to be defined, of an area named the peritumoral brain zone (PBZ), at the margin of the tumor central core [8].

Although the original identification of GSCs dates back more than a decade, the purification and characterization of GSCs remain challenging. Since they play important roles in mediating therapeutic resistance through supporting radio-resistance, chemo-resistance, angiogenesis, invasion and recurrence, we need a deeper understanding of how to selectively target and ablate these tumor-initiating and -propagating populations [9,10,11,12,13]. The most compelling reason to study glioma biology with GSCs is the fact that they have been shown to be very tumorigenic in vivo and form diffuse and invasive tumors that are highly resistant to conventional treatments, indicative of actual patient disease in clinic [14,15].

The GBM-PBZ is a region that radiologically and macroscopically resembles normal brain tissue, but with a particular cellular content, which consists of infiltrating tumor cells, reactive astrocytes, inflammatory cells and other stromal cells [8]. Furthermore, since in 90% of cases tumor recurrence occurs at the margin of the surgical cavity, even after a complete tumor resection and chemo-radiotherapies, the glioma microenvironment seems to be a critical regulator of tumor progression [16]. Therefore, a better understanding of this area is crucial to unravel the mechanisms underlying the GBM relapse and to develop new therapeutic approaches [8,17].

At the DNA level, GBMs are usually characterized by high levels of genomic instability with high rates of copy number alterations (CNAs), easily identifiable by array comparative genomic hybridization (array-CGH) [9,18,19,20]. Thanks to these studies, frequently amplified genes, such as EGFR, MET, PDGFRA, MDM2, PIK3CA, CDK4 and CDK6, and deleted genes such as CDKN2A/B, PTEN and RB1, have been highlighted [21,22]. However, the constant improvement in genetic characterization of GBMs is still failing to be translated to clinical practice, suggesting that other discovery paradigms should be examined.

Considering the importance of CNA data, in this work we performed an in depth study using array-CGH in order to outline the genomic profiles of 20 tumors and 7 peritumoral biopsies; in addition, we compared the genomic profiles of 14 tumor biopsies with their derived tumorspheres to identify new specific biomarkers useful for new therapies.

## 2. Materials and Methods

### 2.1. Study Population

The patient population of this study consists of 20 adults of both sexes diagnosed with high-grade gliomas. The study was approved by the ethic committee “Comitato Etico Monza e Brianza” (study number: 0031436—GLIODRUG-V, approved on 3 January 2020). Patients undergoing a craniotomy for a high-grade glioma were enrolled between January 2020 and September 2021 by the Neurosurgery Unit of the San Gerardo Hospital (Monza Brianza, Italy) after informed consent was signed. The criteria for histological analyses were based on the recommendations of the 2021 WHO classification of CNS tumors [23] and samples from patients without a confirmed high-grade glioma were excluded from the study. Demographic and clinical data are reported in Table 1.

### 2.2. Biopsy Collection

After surgical removal of the tumor with perilesional dissection [24], the surgical cavity was checked with an intraoperative ultrasound (BK Medical, Herlev, Denmark) for tumor remnants. All patients received a resection of at least 95% of contrast enhancing tumor. The majority of the tumor was sent for formal histological diagnosis. When the surgical bed was considered tumor free, a sample of the surgical cavity was taken. The non-neoplastic tumor margin (PBZ) was collected from what was considered far from functionally eloquent areas under neuronavigation (BrainLab, Munich, Germany) and ultrasound guidance (BK Medical, Denmark). Finally, in only 10 out of 20 patients, it was possible to safely collect PBZ samples. When PBZ was collected, a part of the specimen was sent for formal histology in order to check for the absence of tumor remnants. Biopsy samples were placed in cooled PBS (Euroclone S.p.A., Milan, Italy) containing 10% antibiotics (streptomycin/penicillin, Euroclone S.p.A., Milan, Italy) for the isolation of GSCs [25]; small pieces were also stored for DNA extraction.

### 2.3. Immuno-Molecular Analysis

Immuno-molecular analysis was performed according to routine diagnostic procedures. Immunohistochemical staining was performed on FFPE (4% formalin) sections of 1-μm thickness, according to the manufacturer’s protocols, using the automated instrument Dako Omnis (Agilent Technologies, Santa Clara, CA, USA). All antibodies were purchased from Dako. For p53, only nuclear staining was considered positive. MGMT promoter methylation and IDH mutation analysis were performed as previously reported [26].

### 2.4. GSCs Isolation from Tumor Tissues

GSCs were isolated directly from the TC samples after surgery. Briefly, samples obtained from tumor cores were washed with PBS and placed in a Petri dish. Then, they were disaggregated mechanically and enzymatically with a 1X trypsin-EDTA solution (Euroclone S.p.A., Milan, Italy). The digested tissue was passed through a cell strainer (70 μm) and finally subjected to lysis of red blood cells. Lastly, single cell suspension was seeded in a complete neural stem cell (NSC) culture medium (see below) at a density of 40,000 cells/cm^2^.

### 2.5. Primary GSC Cultures Conditions

GSCs were cultured in a selective medium for NSC, composed of DMEM F-12 and Neurobasal 1:1, B-27 supplement without vitamin A (Life Technologies Italia, Milan, Italy), 2 mM L-glutamine (Euroclone S.p.A., Milan, Italy), 10 ng/mL recombinant human bFGF and 20 ng/mL recombinant human EGF (Miltenyi Biotec, Bergish Gladbach, Germany), 20 UI/mL penicillin and 20 µg/mL streptomycin (Euroclone S.p.A., Milan, Italy). After isolation, the medium was replaced every 3 days to remove stroma and red blood cells residues, catabolic products and to supply fresh nutrients. Debris and adherent death cells generally were eliminated after a couple of passages. The isolated cells propagate in culture as free-floating spheres defined as tumorspheres [6], which appeared in 15–20 days of culture after isolation. When tumorspheres reached an average size of 100 μm in diameter, the culture was ready to be passed and expanded. At each passage (P), tumorspheres were mechanically dissociated using a sterilized p200 pipette set at 180–200 µL and pipetting up and down 100–150 times to achieve a single-cell suspension.

### 2.6. Established Glioma Stem Cell Lines

Two established glioma stem cell lines were used as a positive control of stemness. G166 and G179 cell lines were kindly provided by Professor A. Smith of the Wellcome Trust Medical Research Council Stem Cell Institute, University of Cambridge, Cambridge (UK). These cell lines were extensively characterized by Pollard and Baronchelli [9,14]. The established stem cell lines were cultured as the GSC primary cultures; a G166 line grows as floating spheres, otherwise G179 grows in semi-adhesion.

### 2.7. Clonal Assay

Mechanically dissociated tumorspheres were seeded into 96-well plates at a density of 10 cells per mL in culture medium. Colony formation was scored 7–10 days after initial seeding. The self-renewal efficiency or the percentage of cells that formed spheres was determined by the following formula: $n=\frac{Y}{X}\;\cdot \;100$; where *Y* is the number of wells in which one tumorsphere is developed from a single cell and *X* is the number of wells in which a single cell was present [27]. Wells containing either none or more than one cell were excluded from the analysis.

### 2.8. Differentiation Assay

Mechanically dissociated tumorspheres were seeded at a density of 1 × 10^5^ cells/well into 6-well plates, with a coverslip on the bottom of each well, and into culture medium permissive for differentiation without EGF and with 5% FBS (Euroclone S.p.A., Milan, Italy). After 4 days of culture, the medium was replaced with fresh medium with 5% FBS, without growth factors. Under these conditions, the detection of the three neural lineages was evidenced at 7 days after plating by immunofluorescence.

### 2.9. Immunofluorescence

To evaluate the expression of stemness and differentiation markers, the following antibodies were used: anti-CD133 (1:50, Santa Cruz Biotechnology, Dallas, TX, USA); anti-nestin (1:50, Millipore, Burlington, MA, USA,); anti-GFAP (1:200, DakoCytomation, Glostrup, Denmark); anti-βIII Tubulin (1:100, Cell Signaling, Danvers, MA, USA); anti-MBP (1:50, Santa Cruz Biotechnology, Dallas, TX, USA). Each marker was analyzed in a separate set of experiments and with at least two replicates.

### 2.10. DNA Extraction and Purification

DNA was extracted from GSC primary culture pellets (between P4 and P6), from tumor and peritumor biopsies and from patients’ blood (used as reference) using the automatic extractor iPrep TM (Thermo Fisher Scientific, Waltham, MA, USA) and using kits supplied with the instrument: iPrep tissue, for DNA extraction from cell pellet, TC and PBZ; and iPrep whole blood, for DNA extraction from peripheral patients’ blood. Then, DNA was purified using Genomic DNA clean & concentration kit (Zymo Research, Irvine, CA, USA) based on several washings and elution on column in order to obtain DNA ultra-pure. The concentration and the purity of the extracted DNA were determined by measuring the absorbance (A260/280) of the sample with NanoDrop ND-1000 Spectrophotometer (Thermo Fisher Scientific, Waltham, MA, USA). In some cases, the extracted DNA from primary culture was not sufficient to perform the analysis and it was amplified using the GenomePlex Whole Genome Amplification (WGA) Kit (Sigma-Aldrich, St. Louis, MI, USA), according to the manufacturer’s instructions. Amplified DNA was tested for purity and concentration as above.

### 2.11. Array-CGH

Array-CGH analysis was performed using 60-mer oligonucleotide probe technology (SurePrint G3 Human CGH 8 × 60 K, Agilent Technologies, Santa Clara, CA, USA), according to the manufacturer’s instructions. Agilent Feature Extraction was exploited to generate raw data, which were further analyzed using Cytogenomics 5.1 with the ADAM-2 algorithm (Agilent Technologies, Santa Clara, CA, USA). A minimum of three consecutive probes/regions was considered as a filter. The threshold for genomic deletion is x = −1; the threshold for genomic gain is x = +0.58. The estimated percentage of mosaicism was calculated using the formula reported in [9]. Notably, in a mosaic scenario, the threshold is between −1 and 0 for deletions and between 0 and +0.58 for duplications. Amplifications and homozygous deletions are considered with threshold >+2 and <−1, respectively.

### 2.12. Bioinformatics Analysis

The Database for Annotation, Visualization and Integrated Discovery (DAVID), v 6.8 https://david.ncifcrf.gov/summary.jsp/ (accessed on 7 June 2021) [28,29], was used to analyze the lists of genes included in CNAs shared in at least 3 samples. The chart function was used to identify pathways in which genes in gain and in loss are involved. The clustering function was used to cluster the pathways found in groups with their own enrichment score.

### 2.13. Statistical Analysis

A chi-square test was used to compare data relating to patient-derived primary GSC cultures and those obtained from control GSC cell lines. Pearson correlation was used to compare genetic alteration profiles in matched pairs of TCs and patient derived GSCs and TCs and related PBZ samples. The statistical significance for each pair of correlations was calculated by consulting the Table of Critical Values for Pearson’s R. The level of significance for a two-tailed test was set to α = 0.05. The EASE Score, a modified Fisher’s Exact Test, is used by the DAVID bioinformatics program. *p* < 0.05 was considered statistically significant.

## 3. Results

### 3.1. Clinicopathological Characteristics of GBM Patients

Our cohort consisted of 4 female patients and 16 male patients with a mean age at diagnosis of 63 (ranging from 39 to 81 years old). According to the 2021 WHO guidelines [23], 19 tumors had been classified as glioblastoma IDH-wildtype (GBM), one (GP8) as astrocytoma IDH-mutant, grade 4 (ASG4) and one (GP21), as astrocytoma IDH-mutant, grade 3 (ASG3) (Table 1). The immunophenotypic profile (ATRX, Olig-2 and p53) and the hypermethylation status of MGMT promoter, were assessed on FFPE tumor tissue by routine analysis by experienced pathologists (Table 1).

To maximize the ability to correlate our results with clinical features, as the patients were enrolled from January 2020 to September 2021, they were further classified into 3 categories based on the actual follow-up, if available: dead of disease (DOD); progression free (PF), in case of alive patients lacking recurrence; and alive with recurrence (R).

### 3.2. Isolation, Expansion and Characterization of GSC Primary Cultures

We were able to generate and expand 15 primary cell cultures with an isolation efficiency of 75%. About 15–20 days after the isolation, we observed the formation of tumorspheres in 15 out of 20 patient-derived cultures (Figure 1a). However, as the sole formation of tumorspheres does not provide, per se, a demonstration of the presence of GSCs, we evaluated the self-renewal efficiency and multipotency. We performed clonal assay experiments on our GSC primary cultures and their average self-renewal efficiency (78%) was not statistically different from those of two established GSC lines (83%) (chi-square test) (Figure 1b) [14]. The stem nature of our GSC primary cultures was further demonstrated by evaluating the expression of specific markers of stemness (CD133 and nestin). In addition, we also assessed their ability to differentiate in neural lineages by evaluating GFAP, β III tubulin and MBP expression. All the investigated markers were expressed in our GSC primary cultures similarly to the control cell lines (Figure 1c,d).

### 3.3. Genomic Profiles of Tumor Biopsies Confirmed Canonical Alterations of GBM and Inter-Tumor Heterogeneity

We performed a genomic characterization of 20 tumor core biopsies (TCs) by array-CGH. A total of 78 copy number alterations (CNAs) disrupting the three canonical pathways involved in gliomagenesis, p53, Rb and PIK3KC, were found in our samples (Table 2). The mean mutation burden of canonical CNAs was 3.9 per sample, ranging from zero (TC6 and TC15) to even 7–8 (TC13, TC18 and TC27). The most altered pathway, with 35 alterations (~45% of the total canonical CNAs), was the Rb pathway: 17 out of 20 tumor biopsies (85%) had imbalances in this pathway. The 9p21.3 locus (CDKN2A/B) was lost in 12 tumors; 11 biopsies had gains in 7q21.2 (CDK6); 12q14.1 locus (CDK4) was affected in 6 samples (five gains and one loss); 4 samples led to a loss in the 13q14.2 locus (RB1); finally, locus 12p13.32 (CCND2) was altered in two samples. The second pathway with the greatest number of alterations (34, ~44% of the total canonical CNAs) was the PIK3KC pathway: 17 out of 20 tumors (85%) were found to be altered in its genes. The EGFR locus (7p11.2) had gains in 14 biopsies; the PTEN locus (10q23.31) was affected in 14 samples (13 losses and 1 gain); 3 samples had a gain in the PDGFRA locus (4q12), while 2 tumors lost in the NF1 locus (17q11.2); one tumor had a gain in the 5q13.1 locus (PIK3R1), but none had alterations in the 3q26.32 locus (PIK3CA). The p53 pathway was the least affected canonical pathway, but it was also the one with the fewest genes considered: ~11% of the total CNAs in only 9 out of 20 tumors (45%). The MDM2 locus (12q15) had 4 imbalances (3 gain and 1 loss); the TP53 locus (17p13.1) has 2 losses and 1 gain, while MDM4 locus (1q32.1) had 1 loss and 1 gain. Definitely, the most altered canonical loci in our cohort were 7p11.2 (EGFR), 10q23.31 (PTEN), 9p21.3 (CDKN2A/B) and 7q21.2 (CDK6).

Moreover, considering the total burden of genomic alterations, not just in the canonical pathways, the total CNAs’ load was 529, with a median of 23.5 and range 7–87.

Patients with a heavier burden of CNAs (≥23) relapsed (GP11 and GP18) or died of disease (GP7, GP10, GP12, GP23, GP24 and GP27) within a year from the diagnosis. However, patients with a lighter burden of CNAs (<23) relapsed (GP9) or died of disease (GP6, GP13 and GP14) one year after diagnosis, pointing out that other factors contribute to worsening of prognosis, in addition to CNAs (Table 1 and Table 2).

### 3.4. Genomic Profiles of Tumor Biopsies and Matched Derived Tumorspheres Showed a Good Correlation

In this study, we performed a genomic analysis of 15 patient-derived GSCs in order to further demonstrate their stemness properties, i.e., the ability to contain the genomic aberrations typical of GBM and perpetuate them indefinitely. In one case (GSC15) no alterations were reported, probably due to a low enrichment of the stem components in the culture, so it was not included in further analysis (Table 3).

We compared the genomic profiles of 14 GSCs and their relative TCs. Shared imbalances were evaluated for their dimension (base pair length) and their representativeness (mosaicism percentage) (Appendix A). Furthermore, in order to estimate the similarity between the respective pairs of genomic profiles, we calculated the Pearson correlation metric. We divided the patients into three groups, based on the strength of correlation [30]: very strong correlation (Pearson value R ≥ 0.80), moderately strong correlation (0.40 ≤ R < 0.80); and low correlation (R < 0.40) (Table 4).

Seven patients belong to the first group, with Pearson values from 0.80 to 0.94. Three out of 14 patients are part of the second group, with Pearson values from 0.55 to 0.79. In these two groups, shared CNAs concerned mainly the loss of genomic material; numerical imbalances were less represented in TCs, demonstrated by the presence of mosaicism in most cases, while in GSCs the alterations were more homogeneous, thanks to a probable clonal selection. Notably, the number of alterations increased in GSC cultures, with respect to the matched TCs, except GP10, GP12, GP17 and GP18 (Table 2 and Table 3). Patient GP18 relapsed within one year from surgery, they had a Pearson value score of 0.81. Patients GP7, GP10, GP12, GP13, GP23 and GP24 died of disease within a year after diagnosis and their Pearson values were, respectively, 0.94, 0.9, 0.87, 0.93, 0.92 and 0.55. All p values of our correlation metric resulted <0.05, showing a significant and strong correlation between TC and GSC of each patient of these two groups.

Four out of 14 patients were part of the low correlation group, with Pearson values from 0.07 to 0.16. Furthermore, for this group shared CNAs concerned mainly the loss of genomic material, and the imbalances were more represented in GSC cultures than in biopsies. Patient GP6 was the only member of this group to die of disease within a year from the diagnosis, they had a Pearson value score of 0.07. The p value of our correlation metric resulted <0.05 only in patient GP20.

### 3.5. Comparing the Genetic Profiles of Matched TCs and GSCs of Different Patients to Find New Targets for Therapies

We used DAVID (Database for Annotation, Visualization and Integrated Discovery) in order to investigate the pathways of genes involved in “common” aberrations detected in matched TCs and GSCs and shared among at least three different patients. We presented the top ten pathways recognized by the KEGG pathway database and the top ten GO functions (biological process: BP, cellular component: CC, and molecular function: MF) with statistically significant results. Gained genes were mainly involved in biological processes such as mRNA splicing via spliceosome, leukocyte migration and endocytosis. Cytological composition analysis showed that most parts of the genes were significantly involved in the composition of membrane, extracellular region and extracellular space. The molecular functions were mainly concentrated in receptor binding, carbohydrate binding and lipid binding. The KEGG-pathway showed that these genes were mainly involved in the pathways in cancer, the Rap1 signaling pathway, Huntington’s disease and the cAMP signaling pathway (Table 5). Lost genes were mainly involved in biological processes such as oxidation-reduction and the lipid catabolic process. Most parts of these genes were significantly involved in the cytological composition of cytosol, nucleoplasm and mitochondrion. Their molecular functions were mainly involved in oxygen binding and cytoskeletal protein binding. The KEGG-pathway showed that these genes were mainly involved in metabolic pathways, biosynthesis of antibiotics, the WNT signaling pathway and carbon metabolism (Table 6).

Additionally, we evaluated the gene enrichments. Regarding gained genes, 157 genes were enriched in KEGG pathways, 39 in MF, 33 in BP and none in CC. Then, we observed that 42 lost genes were enriched in BP, 24 in MF, 9 in CC and none in KEGG pathways. Subsequently, we generated Venn diagrams for pathways/GO functions, in order to show the shared genes, both in gain and in loss, by the four datasets. The diagram with the gained genes shows that 12, 4 and 3 genes were shared between two datasets: BP and KEGG, BP and MF, KEGG and MF, respectively (Figure 2a). Similarly, the diagram with the lost genes shows that 15 genes were shared between BP and MF, while only one gene was shared in CC and MF (Figure 2b). Furthermore, both diagrams show no gene shared by all datasets simultaneously.

### 3.6. Revelations from the Comparison between Genomic Profiles of Tumor Core Biopsies (TCs) and Peritumoral Brain Zone (PBZs) Samples

In 10 patients it was also possible to collect the non-neoplastic peritumor margin; therefore, we have also extended the genomic analysis on these biopsies. However, three PBZ samples (PBZ6, PBZ9 and PBZ20) did not report any CNA, so, they were not included in further analysis. We compared the genomic profiles of 7 TCs and the matched PBZs and we calculated the Pearson correlation metrics, dividing the patients into three groups, based on the strength of the correlation [31]: very strong correlation (Pearson value R ≥ 0.80), moderately strong correlation (0.40 ≤ R < 0.80); and low correlation (R < 0.40) (Table 7).

GP11 and GP27 had a great correlation, with Pearson values, respectively, of 0.94 and 0.91, both were statistically significant (*p* < 0.05). Patient GP11 relapsed within one year from surgery and GP27 died of disease. GBM canonical alterations already present in TCs were confirmed in the related PBZs, confirming a very good overlapping of the genomic profiles (Appendix A).

Two out of 7 patients, GP10 and GP17, belong to the second group, with Pearson values, respectively, of 0.62 and 0.42, both are statistically significant (*p* < 0.05). In this group, shared CNAs concerned both loss and gain of genomic material. Numerical imbalances were less represented in PBZs, demonstrated by the presence of mosaicism, confirming the infiltration of cancer cells in this area. Interestingly, patient GP10 died of disease within a year after diagnosis and is the patient with the highest number of shared CNAs between TC and PBZ (Appendix A).

Three patients are part of the low correlation group. GP24 died of disease within one year from surgery and had a Pearson value of 0,37. Notably, patient GP15 and GP22 had a negative correlation, with a Pearson value score, respectively, of −0,87 and −0,52 (*p* < 0.05); a negative correlation attests that the two genomic profiles not only are not overlapping, but they are exclusive and unique (Appendix A).

However, it is interesting to pay attention to some imbalances exclusive of PBZs and not shared with other samples of the same patient (Appendix A). In this case, gains seem more numerous than losses. Noteworthy are two gained regions: 11p11.2, evidenced in PBZ of GP9 and GP22; and 16p13.3, evidenced in GP15 and GP27 (Appendix A). Both regions include genes of interest for glioblastoma, such as *EXT2*, *SSTR5*, *SSTR5-AS1*, *C1QTNF8* and *CACNA1H*. Furthermore, two regions in gain (1p34.2 and 2q14.2) were evidenced specifically only in two PBZs of two patients (GP10 and GP27, respectively).

Finally, in three patients (GP10, GP22 and GP24), we evidenced several CNAs shared between PBZ with their matched GSCs and TCs. Curiously, in some cases the alterations were identified only in the PBZ and in the matched GSCs, suggesting that the tumor cells infiltrated in the PBZ may be the GSCs (Appendix A).

## 4. Discussion

GBM is the most common and fatal primary brain tumor with a median survival rate of only 15 months after the first diagnosis. The current standard of care for GBM, as proposed by Stupp in 2017 [2], consists of maximal safe surgical resection, radiotherapy with concomitant temozolomide, followed by adjuvant temozolomide and tumor-treating fields. Unfortunately, despite the treatment, about 70% of these tumors recur with de novo or acquired resistance, which leads to a low five-year survival rate [32,33,34]. The reasons for this high failure rate are different and closely interconnected. First of all, similarly to what happens in benign tumors such as meningiomas, surgery cannot be considered curative, as the presence of cortical and subcortical functional areas, combined with the widespread infiltrative pattern of tumor growth, makes it difficult to perform a complete surgical resection enclosing a wide area of peritumoral brain [35]. As a consequence, a variable number of invading tumor cells are invariably left behind in the peritumoral brain zone (PBZ), which is most often the site of recurrence [16,36]. Secondly, the huge variability of this type of tumor plays a paramount role: interpatient, intratumoral, functional and molecular heterogeneity, widely described in literature, and which is most likely supported and nourished by GSC subpopulations, makes this tumor very resistant to adjuvant treatments that should destroy the neoplastic cells left behind by the surgeon. In light of this, it is urgent to find more effective therapies that target specifically GCS subpopulations, and simultaneously, to develop in vitro models able to reliably recapitulate the original tumor and that can be used for preliminary rapid and cost-effective testing [37,38]. Establishing primary GSC primary cultures would provide a valuable and accurate model of the human tumor, would give insight into the origins of tumor heterogeneity, and finally, would direct towards the most effective therapies for the patient [39]. In our study, we have shown that a simple and not expensive protocol is valid to establish primary cultures enriched with GSCs in at least 70% of cases. Indeed, the characteristics of stemness of our primary cultures, such as the formation of perpetuating tumorspheres in different passages and the ability to differentiate in neural lineages, have been confirmed. One of the ways to understand if the isolated cells faithfully represent the GSC subpopulations is to evaluate the similarity of the genomic profiles with their tumor cores (TCs). Moreover, we have already shown that the tumorspheres had specific genomic profiles, which can be used as a specific tracer of these subpopulations [9].

In the last years, several studies focusing on genomic, transcriptomic and methylomic analyses of GBM were completed [9,20,40,41,42,43]. In addition, Lemée J. et al. also characterized the peritumoral brain zone of GBM, examining its genomic, transcriptomic and proteomic profiles, but it does not include the GSC component [8]. In this work, for the first time to our knowledge, we obtained a complete view of the genomic profiles of GBM, studying, wherever possible, the tumor core, the relative GSCs and also including the PBZ, all derived from the same patient.

First of all, we obtained the genomic profile of 20 TCs, highlighting that our cohort almost completely recapitulates the information reported in literature. In fact, the most frequent copy number alterations were 7p11.2 amplifications (EGFR locus), 9p21.3 deletions (CDKN2A locus) and 10q23.31 deletions (PTEN locus), that are typical features of primary GBM, essential for gliomagenesis [44]. However, tumor-specific profiles in terms of CNAs distribution were also observed, in accordance with the well-known heterogeneity of GBM.

At the same time, we were able to isolate and expand GSC primary cultures from 15 TCs and we compared their genomic profiles with the matched TC. We showed that 14 out of 15 GSC primary cultures very faithfully reflected the genomic profile of their original TC, as evidenced by the Pearson correlation and by the high number of shared CNAs. We observed that losses are better maintained than gains, and that all the alterations identified in TCs become more homogeneous and represented in GSC cultures, thanks to in vitro clonal selection. These shared CNAs are enriched by genes mainly involved in the pathways of cancer, as evidenced by GO and KEGG enrichment analysis, reinforcing the hypothesis that they are important for the progression and maintenance of cancer cells. We evidenced that a very large number of genes involved in gains and shared between TCs and GSCs belong to the GO term cellular component, particularly extracellular space, extracellular region and membrane, indicating a strong involvement in intercellular signaling. Interestingly, several gained genes shared between at least two datasets were reported in the literature as overexpressed in high-grade gliomas, associated with poor prognosis, or in any case, involved in canonical pathways of GBM (*EIF2AK1*, *FTL*, *SRC*, *GNG8*, *GNG11*, *GNG7*, *FZD9*, *GNB2*, *TGM2*) [45,46,47,48,49]. Conversely, only one lost gene, *AKR1C2*, shared between at least two datasets, was reported as downregulated and associated with high-risk-score glioma in the REMBRANDT data set [50]. These genes could represent interesting markers for prognosis and for new therapies in the future (Table 8).

In this study, we were able to draw a complete picture of genomic profiles of GBM in 10 patients, adding information from the PBZ. There are still very few data on this area in the literature, and they are generally based on gene expression or imaging analysis [55,56]. The larger sharing of genomic profiles would testify to a greater infiltration of tumor cells in healthy areas of the brain, which, after surgical removal of the tumor core, would remain in the site, with a high risk of recurrence formation. Our data confirmed a previous study, which reported that tumor cell infiltration was found in one third of PBZs, despite radiological and macroscopic analysis revealing normal brain tissues [8]. However, histopathological examination did not find tumor cell infiltration in our series of PBZ, even if the overlap between PBZ-TC genomic profiles would prove otherwise for more than 50% of cases. Interestingly, we evidenced two exclusive PBZ-CNAs that could identify a specific signature in this specific area, because it was not present in other samples, nor in other patients. The first includes the *SCMH1* gene, associated with the Polycomb group (PcG) multiprotein complexes, required to maintain the transcriptionally repressive state of some genes [57]. The second region includes the *GLI2* gene, which promotes cell proliferation and migration in glioma [58,59]. Other potentially interesting regions are two gains, in 11p11.2 and 16p13.3, shared by two PBZs and evidenced also in other TCs and/or GSCs from other patients. Several genes included in these regions could hide an important role in the progression of glioma, as reported by the literature. For example, the expression level of the *EXT2* gene is increased in glioblastoma [60]; *C1QTNF8* promotes temozolomide resistance [61]; *CACNA1H* promotes GBM cell proliferation and migration [62]. Finally, we noticed that the CNAs in gains are more represented in PBZ samples, compared to the losses. We have already highlighted this phenomenon in other types of tumor [63], associating it with a possible mechanism of endoreduplication-polyploidization [64]. In support of this, the karyotype of GSCs of GP22 is tetraploid (data not shown), reinforcing the importance of this mechanism in glioma tumorigenesis.

## 5. Conclusions

In this study we were able to draw a complete picture of genomic profiles of GBM, from TCs to PBZs, including GSCs. Although the numbers are too few to draw conclusions, it is curious to note how some patients have died (GP24 and GP10) and their TCs, GSCs and PBZs have very similar genomic profiles. We have confirmed some strong points of glioma tumorigenesis, but we have also contributed to the identification of new genes and new regions potentially useful for future therapeutic strategies. The high heterogeneity of GBM requires the enrollment of a high number of patients in order to achieve statistically significant results and to understand the complexity of the disease. The limitations of our study derive from the small number of enrolled patients and the limited follow up. Nevertheless, the study of genomic profiles is a good starting point for a comparative analysis among the different components of GBM, and this is a strong point of our work. In the future we will move to high throughput approaches, such as single-cell analysis, in order to have a more complete overview of the disease.

## Figures and Tables

**Figure 1 biology-10-01157-f001:**
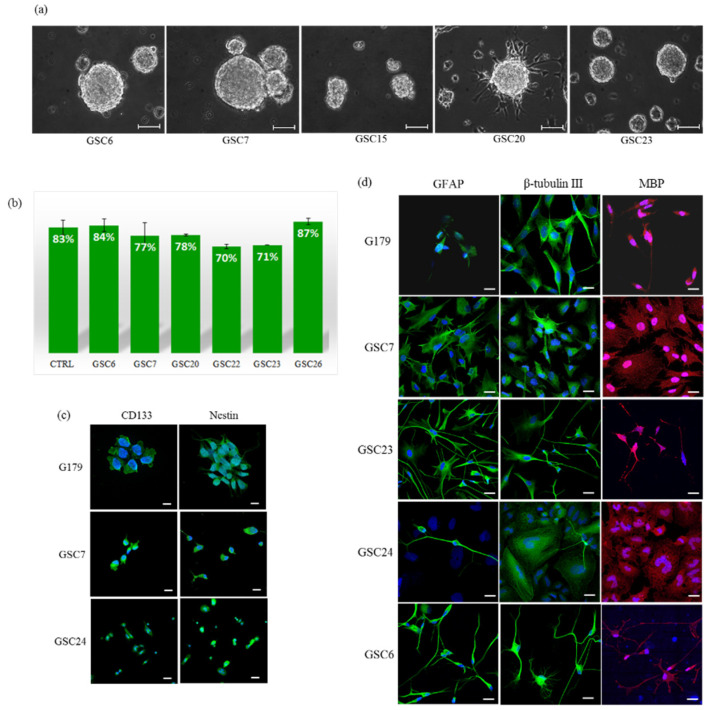
GSC primary cultures: (**a**) Representative images of glioblastoma patient primary cultures (GSCs) growth as floating or semi-adherent tumorspheres. Images were captured at low-passages (P4 and P5) with a Leica DFC290 Microscope Camera at magnification 20X. Unit of measurement bar = 100 μm. (**b**) Self-renewal efficiency. Data shown are the mean ± SEM of two tests in triplicate for each culture. CTRL is the average of the positive controls (G166 and G179 stem cell lines). (**c**) Evaluation of stemness, and (**d**) differentiation markers. A G179 established glioma stem cell line was used as a positive control. Markers can be appreciated in green and red, while nuclei are stained in blue (DAPI). The images refer to areas particularly enriched with cells positive for the specific marker. Unit of measurement bar = 20 μm.

**Figure 2 biology-10-01157-f002:**
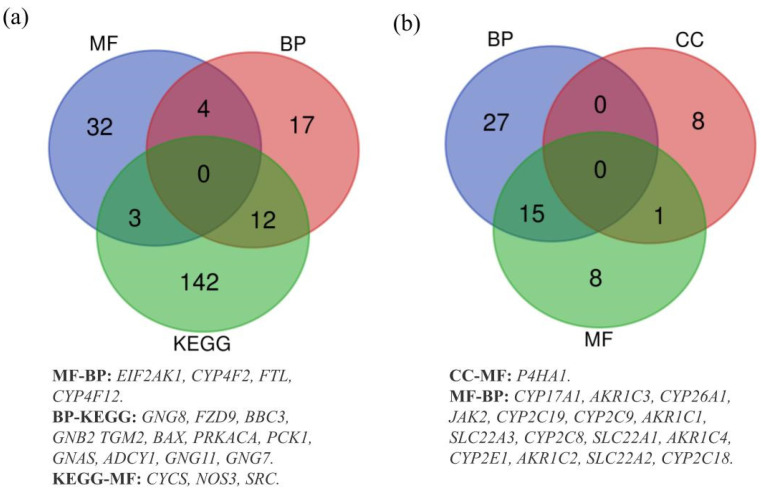
(**a**) Overlapping gained genes in KEGG, BP and MF datasets. (**b**) Overlapping lost genes in BP, CC and MF datasets.

**Table 1 biology-10-01157-t001:** Clinicopathological characteristics and immunomolecular phenotype of glioma patients (GP) enrolled in the study. M: male; F: female; GBM: glioblastoma IDH-wildtype; ASG4: astrocytoma IDH-mutant, grade 4; ASG3: astrocytoma IDH-mutant, grade 3; wt: wild-type; mut: mutated; +: <30% of positive cells; -: no staining; met: methylated; follow-up period expressed in months; DOD: patient dead of disease; R: relapsed patient, follow up ongoing; PF: progression free patient, follow up ongoing; n.a.: no information available.

Patient ID	Sex	Age (Years)	Diagnosis	Immunomolecular Phenotype	Follow-Up
GP5	M	57	GBM	IDH wt, ATRX+, p53-, MGMT wt	n.a.
GP6	M	44	GBM	IDH wt, ATRX+, p53-, MGMT wt	15, DOD
GP7	M	54	GBM	IDH wt, ATRX+, p53-, MGMT wt	15, DOD
GP8	M	57	ASG4	IDH mut, ATRX+, p53-, MGMT met	15, PF
GP9	M	71	GBM	IDH wt, ATRX+, p53-, MGMT met	15, R
GP10	M	74	GBM	IDH wt, ATRX+, p53-, MGMT wt	3, DOD
GP11	M	76	GBM	IDH wt, ATRX+, p53-, MGMT met	15, R
GP12	F	70	GBM	IDH wt, ATRX+, p53-, MGMT wt	3, DOD
GP13	M	58	GBM	IDH wt, ATRX+, p53-, MGMT met	13, DOD
GP14	M	73	GBM	IDH wt, ATRX+, p53-, MGMT met	1, DOD
GP15	M	78	GBM	IDH wt, ATRX+, p53-, MGMT met	11, PF
GP17	M	81	GBM	IDH wt, ATRX+, p53-, MGMT wt	11, PF
GP18	F	49	GBM	IDH wt, ATRX+, p53-, MGMT wt	11, R
GP20	F	60	GBM	IDH wt, ATRX+, p53-, MGMT wt	10, PF
GP21	M	39	ASG3	IDH mut, ATRX-, p53+, MGMT met	8, PF
GP22	M	58	GBM	IDH wt, ATRX+, p53-, MGMT met	8, PF
GP23	M	71	GBM	IDH wt, ATRX+, p53-, MGMT met	6, DOD
GP24	M	69	GBM	IDH wt, ATRX+, p53-, MGMT wt	6, DOD
GP26	F	69	GBM	IDH wt, ATRX+, p53-, MGMT met	4, PF
GP27	M	57	GBM	IDH wt, ATRX+, p53-, MGMT wt	2, DOD

**Table 2 biology-10-01157-t002:** Summary of copy number alterations affecting the three main canonical pathways altered in high-grade gliomas. The analysis was conducted on patient tumor core biopsies (TCs). The canonical CNAs load (Canonical CNA) and the total burden of imbalances (Total CNA) per sample were also reported.

Sample ID	p53 Pathway			Rb Pathway					PIK3KC Pathway						Canonical CNA	Total CNA
	TP53 17p13.1	MDM2 12q15	MDM4 1q32.1	RB1 13q14.2	CDK4 12q14.1	CDK6 7q21.2	CCND2 12p13.32	CDKN2A/B 9p21.3	PIK3CA 3q26.32	PIK3R1 5q13.1	PTEN 10q23.31	EGFR 7p11.2	PDGFRA 4q12	NF1 17q11.2		
TC5					GAIN						LOSS	GAIN			3	52
TC6															0	11
TC7						GAIN		LOSS			LOSS	GAIN			4	27
TC8				LOSS							LOSS				2	19
TC9								LOSS			LOSS	GAIN			3	16
TC10		GAIN			GAIN			LOSS					GAIN		4	55
TC11								LOSS			LOSS	GAIN			3	27
TC12						GAIN		LOSS			LOSS	GAIN			4	27
TC13				LOSS	GAIN	GAIN		LOSS			LOSS	GAIN	GAIN		7	14
TC14						GAIN		LOSS			LOSS	GAIN			4	13
TC15															0	18
TC17	LOSS					GAIN		LOSS		GAIN		GAIN		LOSS	6	24
TC18	LOSS		LOSS			GAIN	LOSS	LOSS			LOSS	GAIN		LOSS	8	87
TC20	GAIN					GAIN						GAIN			3	19
TC21		LOSS			LOSS		GAIN								3	7
TC22		GAIN									GAIN				2	8
TC23				LOSS		GAIN		LOSS			LOSS	GAIN			5	28
TC24		GAIN		LOSS	GAIN	GAIN					LOSS	GAIN			6	23
TC26						GAIN		LOSS			LOSS	GAIN			4	27
TC27			GAIN		GAIN	GAIN		LOSS			LOSS	GAIN	GAIN		7	27
Total	3	4	2	4	6	11	2	12	0	1	14	14	3	2	78	529

**Table 3 biology-10-01157-t003:** Summary of copy number alterations affecting the three main canonical pathways altered in high-grade gliomas. The analysis was conducted on patient-derived primary glioma stem cell cultures (GSCs). The canonical CNAs load (Canonical CNA) and the total burden of imbalances (Total CNA) per sample were also reported.

Sample ID	p53 Pathway			Rb Pathway					PIK3KC Pathway						Canonical CNA	Total CNA
	TP53 17p13.1	MDM2 12q15	MDM4 1q32.1	RB1 13q14.2	CDK4 12q14.1	CDK6 7q21.2	CCND2 12p13.32	CDKN2A/B 9p21.3	PIK3CA 3q26.32	PIK3R1 5q13.1	PTEN 10q23.31	EGFR 7p11.2	PDGFRA 4q12	NF1 17q11.2		
GSC6				LOSS		GAIN		LOSS			LOSS	GAIN			5	57
GSC7	LOSS					GAIN		LOSS			LOSS	GAIN			5	33
GSC8		LOSS			GAIN/LOSS					LOSS	LOSS	LOSS			5	28
GSC10		GAIN			GAIN			LOSS					GAIN		4	53
GSC12						GAIN		LOSS			LOSS	GAIN			4	23
GSC13					GAIN	GAIN		LOSS			LOSS	GAIN	GAIN		6	16
GSC15															0	0
GSC17	GAIN/LOSS					GAIN		LOSS		GAIN		GAIN		LOSS	6	20
GSC18						GAIN		LOSS			LOSS	GAIN			4	18
GSC20						GAIN		LOSS			LOSS	GAIN	GAIN		5	33
GSC21		LOSS			LOSS		GAIN	LOSS							4	25
GSC22		GAIN			GAIN	GAIN				GAIN	LOSS	GAIN			6	49
GSC23			GAIN			GAIN		LOSS			LOSS	GAIN			5	40
GSC24					GAIN			LOSS			LOSS				3	30
GSC26						GAIN		LOSS			LOSS	GAIN			4	34
Total	2	4	1	1	6	10	1	12	0	3	11	11	3	1	66	459

**Table 4 biology-10-01157-t004:** Report of Pearson correlation values (R) for TC-GSC-genomic profiles comparison. GPs are ranked into three groups: ones with a very strong correlation between the pair of genomic profiles (R ≥ 0.80); ones with a moderately strong correlation (0.40 ≤ R < 0.80); ones with a low correlation (R < 0.40). The statistical significance for each pair of correlations was calculated consulting the Table of Critical Values for Pearson’s R. N: total number of CNAs; DF: degree of freedom (N-2). Level of significance for a two-tailed test α = 0.05. * indicates statistically significant test.

		GP	R	N	DF	Critical Value of R
Very strong		7	0.94	32	30	0.349 *
		10	0.9	57	55	0.261 *
		12	0.87	33	31	0.349 *
		13	0.93	18	16	0.468 *
		17	0.87	25	23	0.396 *
		18	0.81	102	100	0.195 *
		23	0.92	45	43	0.288 *
Moderately strong		21	0.65	25	23	0.396 *
		24	0.55	43	41	0.304 *
		26	0.79	37	35	0.325 *
Low		6	0.07	60	58	0.25
		8	0.14	49	47	0.288
		20	0.3	45	43	0.288 *
		22	0.16	57	55	0.261

**Table 5 biology-10-01157-t005:** Enrichment analysis of the GO and KEGG pathways of genes involved in gains.

Category	ID	Term	Count	*p*-Value
BP	GO:2000117	Negative regulation of cysteine-type endopeptidase activity	11	0.0000029
BP	GO:0006298	Mismatch repair	13	0.00061
BP	GO:0001580	Detection of chemical stimulus involved in sensory perception of bitter taste	14	0.00065
BP	GO:0050900	Leukocyte migration	29	0.00068
BP	GO:0006690	Icosanoid metabolic process	6	0.0011
BP	GO:0000398	mRNA splicing, via spliceosome	43	0.0026
BP	GO:0016079	Synaptic vesicle exocytosis	9	0.0033
BP	GO:0001541	Ovarian follicle development	13	0.0036
BP	GO:0006897	Endocytosis	29	0.0052
BP	GO:0045742	Positive regulation of epidermal growth factor receptor signaling pathway	9	0.006
CC	GO:0005615	Extracellular space	211	0.00006
CC	GO:0002080	Acrosomal membrane	10	0.00017
CC	GO:0005576	Extracellular region	242	0.00022
CC	GO:0005681	Spliceosomal complex	22	0.0039
CC	GO:0016020	Membrane	306	0.0052
CC	GO:0042995	Cell projection	18	0.0076
CC	GO:0005834	Heterotrimeric G-protein complex	10	0.014
CC	GO:0097060	Synaptic membrane	7	0.016
CC	GO:0005665	DNA-directed RNA polymerase II, core complex	7	0.016
CC	GO:0032389	MutLalpha complex	5	0.017
MF	GO:0016705	Oxidoreductase activity with incorporation or reduction of molecular oxygen	20	0.000022
MF	GO:0008236	Serine-type peptidase activity	19	0.00033
MF	GO:0004869	Cysteine-type endopeptidase inhibitor activity	13	0.00039
MF	GO:0008327	Methyl-CpG binding	10	0.0004
MF	GO:0030246	Carbohydrate binding	41	0.00054
MF	GO:0070330	Aromatase activity	11	0.00076
MF	GO:0004497	Monooxygenase activity	17	0.001
MF	GO:0005102	Receptor binding	63	0.0014
MF	GO:0008417	Fucosyltransferase activity	6	0.0021
MF	GO:0008289	Lipid binding	31	0.0038
KEGG	hsa04380	Osteoclast differentiation	32	0.00013
KEGG	hsa04911	Insulin secretion	22	0.00083
KEGG	hsa04725	Cholinergic synapse	25	0.0027
KEGG	hsa04932	Non-alcoholic fatty liver disease (NAFLD)	30	0.0064
KEGG	hsa05016	Huntington’s disease	36	0.0069
KEGG	hsa04015	Rap1 signaling pathway	38	0.0097
KEGG	hsa04024	cAMP signaling pathway	3	0.011
KEGG	hsa05030	Cocaine addiction	13	0.011
KEGG	hsa04970	Salivary secretion	19	0.012
KEGG	hsa05200	Pathways in cancer	63	0.013

**Table 6 biology-10-01157-t006:** Enrichment analysis of GO and KEGG pathway of genes involved in losses.

Category	ID	Term	Count	*p*-Value
BP	GO:0071395	Cellular response to jasmonic acid stimulus	4	0.00031
BP	GO:0055114	Oxidation-reduction process	45	0.00033
BP	GO:0016042	Lipid catabolic process	12	0.0011
BP	GO:0044241	Lipid digestion	5	0.0018
BP	GO:0008202	Steroid metabolic process	8	0.0023
BP	GO:0015872	Dopamine transport	4	0.0025
BP	GO:0006096	Glycolytic process	7	0.0031
BP	GO:0044597	Daunorubicin metabolic process	4	0.0038
BP	GO:0044598	Doxorubicin metabolic process	4	0.0038
BP	GO:0071243	Cellular response to arsenic-containing	4	0.0056
CC	GO:0005829	Cytosol	169	0.0042
CC	GO:0005739	Mitochondrion	76	0.0055
CC	GO:0005720	Nuclear heterochromatin	5	0.012
CC	GO:1990246	Uniplex complex	3	0.016
CC	GO:0045334	Clathrin-coated endocytic vesicle	4	0.023
CC	GO:0005761	Mitochondrial ribosome	5	0.028
CC	GO:0042599	Lamellar body	3	0.042
CC	GO:0005604	Basement membrane	8	0.048
CC	GO:0005654	Nucleoplasm	135	0.048
CC	GO:0005922	Connexon complex	4	0.056
MF	GO:0018636	Phenanthrene	4	0.00032
MF	GO:0000987	Core promoter proximal region sequence-specific DNA binding	6	0.00032
MF	GO:0031406	Carboxylic acid binding	4	0.0025
MF	GO:0008392	Arachidonic acid epoxygenase activity	5	0.0033
MF	GO:0016655	Oxidoreductase activity, acting on NAD(P)H, quinone as acceptor	4	0.0039
MF	GO:0019825	Oxygen binding	8	0.004
MF	GO:000809	Cytoskeletal protein binding	8	0.0045
MF	GO:0008395	Teroid hydroxylase activity	6	0.0047
MF	GO:0004806	Triglyceride lipase activity	5	0.0054
MF	GO:0047086	Ketosteroid monooxygenase activity	3	0.0055
KEGG	hsa01200	Carbon metabolism	13	0.0029
KEGG	hsa04310	Wnt signaling pathway	14	0.0056
KEGG	hsa01100	Metabolic pathways	69	0.0082
KEGG	hsa01130	Biosynthesis of antibiotics	17	0.018
KEGG	hsa01230	Biosynthesis of amino acids	8	0.032
KEGG	hsa00591	Linoleic acid metabolism	5	0.033
KEGG	hsa00561	Glycerolipid metabolism	7	0.035
KEGG	hsa00590	Arachidonic acid metabolism	7	0.044
KEGG	hsa04114	Oocyte meiosis	10	0.046
KEGG	hsa01212	Fatty acid metabolism	6	0.051

**Table 7 biology-10-01157-t007:** Report of Pearson correlation values (R) for TC-PBZ-genomic profiles comparison. GPs are ranked into three groups: ones with a very strong correlation between the pair of genomic profiles (R ≥ 0.80); ones with a moderately strong correlation (0.40 ≤ R < 0.80); ones with a low correlation (R < 0.40). The statistical significance for each pair of correlations was calculated consulting the Table of Critical Values for Pearson’s R. N: total number of CNAs; DF: degree of freedom (N-2). Level of significance for a two-tailed test α = 0.05. * indicates statistically significant test.

	GP	R	N	DF	Critical Value of R
Very strong	11	0.94	35	33	0.325 *
	27	0.91	35	33	0.325 *
Moderately strong	10	0.62	71	69	0.232 *
	17	0.42	24	22	0.404 *
Low	15	−0.87	24	22	0.404 *
	22	−0.52	23	21	0.413 *
	24	0.37	36	34	0.325 *

**Table 8 biology-10-01157-t008:** Summary of relevant genes shared at least between two datasets of the enrichment analysis with DAVID. The following is reported: the cytogenetic localization, the function (GeneCards: The Human Gene Database, https://www.genecards.org/, accessed on 20 September 2021) [51], the association with GBM and possible new targeted therapy (DrugBank database; https://go.drugbank.com/, accessed on 20 September 2021) [52].

Gene	Cytogenetic Band	Function	In GBM	Target Drug
EIF2AK1	7p22.1	The protein encoded by this gene acts at the level of translation initiation to downregulate protein synthesis in response to stress.	It is over expressed in glioma tissue rather than in normal brain tissue. Its expression is not variable among different GBM subtype [45].	
FTL	19q13.33	This gene encodes the light subunit of the ferritin protein.	Its expression is enriched in high-grade glioma (HGG) and together with IDH1/2 wildtype it is significantly associated with an unfavorable prognosis of glioma patients [46].	Ferumoxytol (phase 2 clinical trials)
SRC	20q11.23	This proto-oncogene is part of the EGFR/SRC/ERK pathway. It may play a role in the regulation of embryonic development and cell growth.	Its role in YTHDF2 phosphorylation leads to an increase proliferation, invasiveness and tumorigenesis in GBM [53].	Dasinatib (phase 2 clinical trials)
GNG7	19p13.3	A member of the guanine nucleotide-binding proteins (G proteins) gamma family that is involved as a modulator or transducer in various transmembrane signaling systems.	It has been associated with GBM due to their genomic localization close to WNT5A and WNT10B genes whose pathway is known to be altered in GBM [47].	
GNG8	19q13.32			
GNG11	7q21.3			
GNB2	7q22.1	This gene encodes heterotrimeric guanine nucleotide-binding proteins (G proteins), which integrate signals between receptors and effector proteins, and are composed of an alpha, a beta and a gamma subunit.	It was observed that GBM patients with a high GNB2 level are associated with higher pathological grade, wild-type IDH and unmethylated MGMT promoter [48].	
TGM2	20q11.23	Transglutaminases are enzymes that catalyze the crosslinking of proteins by epsilon-gamma glutamic lysine isopeptide bonds.	Its high expression in glioma tissues has been reported [49].	
FZD9	7q11.23	Members of the ‘frizzled’ gene family encode 7-transmembrane domain proteins that are receptors for Wnt signaling proteins.	It is expressed in the brain and its levels are significantly higher in malignant astrocytomas than in low-grade astrocytomas [54].	
AKR1C2	10p15.1	This gene encodes a member of the aldo/keto reductase superfamily.	It was reported as downregulated and associated with high-risk-score glioma [50].	

## Data Availability

The data used to support the findings of this study are available from the corresponding author upon request.

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
