# Peer review of "Characterizing the Genomic Profile in High-Grade Gliomas: From Tumor Core to Peritumoral Brain Zone, Passing through Glioma-Derived Tumorspheres"

_biology, 2021, doi:10.3390/biology10111157_

Round 1

Reviewer 1 Report

The authors were able to clarify each point I made in my review. I just have an additional one: 

  • Conclusion: The authors should use the term 'limited' instead of 'not very long'.

In conclusion, the manuscript ist acceptable in its current form after clarifying the minor point mentioned above.

Author Response

Reviewer 1 Comments and Suggestions for Authors

The authors were able to clarify each point I made in my review. I just have an additional one: 

  • Conclusion: The authors should use the term “limited”; instead of “not very long”.

R: We thank a lot the reviewer for appreciating our work. We corrected the term as suggested (line 553).

Reviewer 2 Report

In this manuscript, M. Giambra and coworkers investigated by using array-CGH platform, the genomic profile of high grade gliomas (mostly glioblastoma IDH-wildtype) from tumour core biopsies (TC), their corresponding peritumoral brain zones (PBZ) and the derived glioma stem cells (GSC) grown as tumorspheres. They obtained a complete view of the genomic profiles of 20 tumour biopsies and compared these data to the genomic profiles of 14 related GSC cultures and to those of 7 PBZ biopsies. Then, the authors investigated by bioinformatics analysis (DAVID database) the pathways of genes involved in the detected genomic alterations and identified new genes and new regions potentially interesting in the glioblastoma study landscape.

This study is well conducted and the answers made by the authors following the reviewers’ comments are appropriate.

Nevertheless, as already highlighted by the previous reviewers, I find myself that too much conclusions are drawn throughout the manuscript from the survival of the patients without any survival analysis. Even if the authors made a substantial effort to moderate them in the present version, these conclusions cannot be allowed regarding the too short follow-up of the series and the small number of enrolled patients. To my opinion, the data reported here should be limited to the genomic profiles and the pathway database investigations which are worth to be published, without any speculative correlation to the clinical status of the patients. Accordingly, Figure S1 should be removed.

Moreover, regarding the glioma patients involved and according to the cIMPACT-NOW consortium (Brat et al., 2020 doi.org/10.1007/s00401-020-02127-9) and the 2021 WHO CNS tumour classification, some terms should be corrected:

  • Glioblastoma multiforme terminology (lines 21, 36, 40, 220) should no longer be used (it was already abandoned in the 2016 WHO version).
  • Glioblastoma without mutation of IDH gene are called now: “Glioblastoma IDH-wildtype”
  • Glioblastoma with IDH mutation (GP8) are called now “Astrocytoma IDH-mutant, grade 4”, with Arabic numeral and not Roman. Glioblastoma IDH-mutant terminology is no longer accepted.
  • For Astrocytoma IDH-mutant grade 3 (GP21), the grade must also be written with Arabic numeral.

Finally, in order to highlight in the paper the relevant data, I would suggest adding Figures 2 and 4 as Supplementary data and putting Table S5 in the main document. I think that it would emphasize the druggable genes identified, which is the main goal of the study.

Author Response

Reviewer 2 Comments and Suggestions for Authors

In this manuscript, M. Giambra and coworkers investigated by using array-CGH platform, the genomic profile of high grade gliomas (mostly glioblastoma IDH-wildtype) from tumour core biopsies (TC), their

corresponding peritumoral brain zones (PBZ) and the derived glioma stem cells (GSC) grown as

tumorspheres. They obtained a complete view of the genomic profiles of 20 tumour biopsies and compared

these data to the genomic profiles of 14 related GSC cultures and to those of 7 PBZ biopsies. Then, the

authors investigated by bioinformatics analysis (DAVID database) the pathways of genes involved in the

detected genomic alterations and identified new genes and new regions potentially interesting in the

glioblastoma study landscape.

This study is well conducted and the answers made by the authors following the reviewers’ comments are

appropriate.

  • Nevertheless, as already highlighted by the previous reviewers, I find myself that too much conclusions are drawn throughout the manuscript from the survival of the patients without any survival analysis. Even if the authors made a substantial effort to moderate them in the present version, these conclusions cannot be allowed regarding the too short follow-up of the series and the small number of enrolled patients. To my opinion, the data reported here should be limited to the genomic profiles and the pathway database investigations which are worth to be published, without any speculative correlation to the clinical status of the patients. Accordingly, Figure S1 should be removed.

R: We thank the reviewer for the criticism. We understood that our speculative data about patients’ survival is not supported by an appropriate analysis due to the limited follow up. We maintained only the follow up record of each patient, removing any speculation from the whole manuscript. We also removed Figure S1 as suggested.

Moreover, regarding the glioma patients involved and according to the cIMPACT-NOW consortium (Brat et al., 2020 doi.org/10.1007/s00401-020-02127-9) and the 2021 WHO CNS tumour classification, some terms should be corrected:

  • Glioblastoma multiforme terminology (lines 21, 36, 40, 220) should no longer be used (it was already abandoned in the 2016 WHO version).

R: We corrected it.

  • Glioblastoma without mutation of IDH gene are called now: “Glioblastoma IDH-wildtype”.

R: We corrected it.

  • Glioblastoma with IDH mutation (GP8) are called now “Astrocytoma IDH-mutant, grade 4”, with Arabic numeral and not Roman. Glioblastoma IDH-mutant terminology is no longer accepted.

R: We corrected it.

  • For Astrocytoma IDH-mutant grade 3 (GP21), the grade must also be written with Arabic numeral.

R: We corrected it.

  • Finally, in order to highlight in the paper the relevant data, I would suggest adding Figures 2 and 4 as Supplementary data and putting Table S5 in the main document. I think that it would emphasize the druggable genes identified, which is the main goal of the study.

R: Thanking the reviewer for the suggestion, we made this change.

Reviewer 3 Report

Dear Authors,

Your manuscript describing the genomic profile of glioblastoma (GBM) samples and tumor sphere cultures derived of them is interesting approach to dissect tumor heterogeneity. Your collection is an impressive collection of matched samples, which is crucial for further understanding of glioma biology. Nevertheless I do have some points that could be improved:

  1. What do you refer to “immunomolecular phenotype” in table 1? Is this data only achieved by immunohistology? Please add the method used to the material and methods section. Also present and not present in this table is very misleading. The histology markers described also rely on position of the staining to be meaningful for diagnosis. I would highly recommend to add sequencing data of at least the genes you are working with, as the Knudson two-hit hypothesis is essential for most of the losses described.
  2. Your collection of tumors does not match the title of the manuscript. You include an astrocytoma grade 3 and an IDH-mutant GBM. Both are by definition (WHO 2021) no GBM. So I would recommend to change the title or exclude these two samples.
  3. Another reason for the exclusion of these two samples is the already high heterogeneity of your collection. Based on different classification approaches there are several GBM subclasses that are biological meaningful (e.g. mesenchymal) and they are defined by certain mutations and chromosomal aberrations. This point is not addressed well in your manuscript. Especially the methylation classification is getting a lot attention, also by the WHO, therefore I highly recommend to refer to the classes described there. In addition the technology that is used to obtain the methylation profile, also enables the user to get very detailed copy number profiles which are comparable to your data.
  4. The processes you use to generate figure 1 should be described in more detail, as the shown pictures raise several questions.
    1. Why do the IF pictures in c) not resemble the spheroids shown in a)?
    2. What is shown in d)? These cells are adherent growing and show different morphology in each picture of one culture. Based on your described protocol for differentiation you should get all cell types in one culture and not such clearly separated cells.
    3. Please check the specificity of your MBP antibody! MBP is a structure protein that should be present only in the cytoplasm and spare the nucleus. In your pictures it is clearly nuclear.
  5. In line 296-299 you state that culture GSC15 is low on stem components, which is the reason for the absent copy number alterations. For me this raises the question, what you think the culture is composed of? Based in your data the culture matched the TC in respect to copy number alterations (figure 2). In addition cells with no stem cell potential should not be able to grow as spheroids in the media you use.

Author Response

Reviewer 3 Comments and Suggestions for Authors

Dear Authors,

Your manuscript describing the genomic profile of glioblastoma (GBM) samples and tumor sphere cultures

derived of them is interesting approach to dissect tumor heterogeneity. Your collection is an impressive

collection of matched samples, which is crucial for further understanding of glioma biology. Nevertheless, I

do have some points that could be improved:

  • What do you refer to “immunomolecular phenotype” in table 1? Is this data only achieved by immunohistology? Please add the method used to the material and methods section. Also present and not present in this table is very misleading. The histology markers described also rely on position of the staining to be meaningful for diagnosis. I would highly recommend to add sequencing data of at least the genes you are working with, as the Knudson two-hit hypothesis is essential for most of the losses described.

R: In Table 1, we referred to “immunomolecular phenotype” to indicate immunohistochemistry and molecular biology routine examinations performed by the diagnostic laboratory, as a standard procedure in the clinic. In particular, immunochemistry analysis is performed to assess the positivity to ATRX, p53 and IDH1. IDH mutational state is even evaluated with PCR analysis. The methylation state of MGMT promoter is analysed with PCR-pirosequencing methodology. We emphasize that they are all preliminary analyses to the study described here, and the immunomolecular phenotype constitute the selection criteria of the enrolled patients. Anyway, the necessary information has been added in the paragraph 2.3.

  • Your collection of tumors does not match the title of the manuscript. You include an astrocytoma grade 3 and an IDH-mutant GBM. Both are by definition (WHO 2021) no GBM. So I would recommend to change the title or exclude these two samples.

R: We thank the reviewer for the suggestion. We corrected the title of our manuscript.

  • Another reason for the exclusion of these two samples is the already high heterogeneity of your collection. Based on different classification approaches there are several GBM subclasses that are biological meaningful (e.g. mesenchymal) and they are defined by certain mutations and chromosomal aberrations. This point is not addressed well in your manuscript. Especially the methylation classification is getting a lot attention, also by the WHO, therefore I highly recommend to refer to the classes described there. In addition the technology that is used to obtain the methylation profile, also enables the user to get very detailed copy number profiles which are comparable to your data.

R: In our work, we did not divide our patients into different GBM subtypes but we preferred to consider each sample individually, for two main reasons. Firstly, our patient cohort is small and therefore we could not reach a significant number of samples for each subclass. Secondly, as Glioblastoma is a highly heterogeneous disease both from an inter-tumor and intra-tumor point of view, our goal was to evaluate the genomic profiles between biopsies and cells from the same patient, therefore it is outside the classifications of the subtypes. Furthermore, in literature it has been reported that most patients display different GB subtypes within the same tumor (Sottoriva A. et al. 2013, Lee E. et al. 2018, Hu LS. et al. 2020).

  • The processes you use to generate figure 1 should be described in more detail, as the shown pictures raise several questions.
    • Why do the IF pictures in c) not resemble the spheroids shown in a)?

R: In Figure 1c) our aim was to evaluate the expression of stemness markers by the cell line control and by our primary cell cultures. Therefore, we performed IF experiments on the second day after cell plating, and we did not wait the formation of spheroids, which could take a variable time, depending on the specific cell culture. Furthermore, the spheroid formation rate is different for every line/culture, e.g. in G179 it is possible to note a principle of sphere formation still absent in GSC7 and GSC24. The spheroid pictures in a) were acquired 10-15 days after plating, depending on the sample, in order to show different examples.

  • What is shown in d)? These cells are adherent growing and show different morphology in each picture of one culture. Based on your described protocol for differentiation you should get all cell types in one culture and not such clearly separated cells.

R: Pictures in d shown line/cultures that underwent a differentiation process. During the differentiation protocol we added 5% of FBS and we gradually removed the growth factors from the culture medium, leading to the adhesion and differentiation of the cells. Each picture of one line/culture shows cells that followed a specific differentiation course, it is normal that they presented different morphologies. For each line/culture, we plated cells on 9 slides (3 slides for each marker). All the plated slides were cultured under the same conditions. Seven days after plating, we started the IF protocol, adding one out of the three differentiated lineage antibodies to 3 random slides. Precisely, for each line/culture, we incubated two slides with anti-GFAP (one slide used as negative control), two slides with anti-MBP (one slide used as negative control) and two slides with anti-βIII Tubulin (one slide used as negative control). In each slides we observed positive cells to a specific marker, but the intrinsic heterogeneity of these samples does not allow for 100% positive cells. We acquired the most representative images for each marker in each line/culture.

  • Please check the specificity of your MBP antibody! MBP is a structure protein that should be present only in the cytoplasm and spare the nucleus. In your pictures it is clearly nuclear.

R: We had already noticed that the protein localization was not canonical and we proved the specificity of our MBP antibody through other experiments. Furthermore, it has been shown that MBP has a variable localization in oligodendrocytes: in some cells is concentrated at the plasmalemma level, while in others is distributed between cytoplasm and nucleus (Hardy RJ. et al.1996; Cytoplasmic and nuclear localization of myelin basic proteins reveals heterogeneity among oligodendrocytes. Journal of Neuroscience Research).

  • In line 296-299 you state that culture GSC15 is low on stem components, which is the reason for the absent copy number alterations. For me this raises the question, what you think the culture is composed of? Based in your data the culture matched the TC in respect to copy number alterations (figure 2). In addition cells with no stem cell potential should not be able to grow as spheroids in the media you use.

R: After surgery, only 5% of the removed tumor was used for research issues, the majority of the sample was used for formal report and diagnosis. In the whole tumor mass the percentage of glioma stem cells is 1-2% (Dick J. E. et al 2008). Different cell components constitute the rest of the glioma mass, such as differentiated tumor cell sub-clones, immune infiltrate, reactive astrocytes, vascular cells, stromal cells, stamness progenitors (transit cells)(Perus L.J.M. and Walsh A.L. 2019). In this work we applied an isolation protocol in order to isolate the stem cell component from a very little portion of biopsy. The stemness nature needs to be demonstrated by the evaluation of self-renewal capability, multipotency and genomic profile studying, since growing as neurospheres in a stem medium is not sufficient, as in the case of GSC15. We supposed that this culture contained a very low percentage of stem cells (<1%) because it didn’t show the characteristic CNAs and it had a limited number of passages in culture. Nevertheless, stem cells are not the only cellular component that can grow as spheroids in stemness medium for a limit number of passages, transit cells can do the same.

Round 2

Reviewer 3 Report

Dear Authors,

Thank you for addressing all remarks adequately and explaining them. Now there is only one point that still puzzles me. Based on your explanation how figure 1 (d) was generated one would expect all kind of cells per well. But in your figure it looks like nearly all cells always follow the same differentiation path, as the whole picture is filled with positive cells. I would expect all pictures to look more like the GSC24 GFAP staining. Some cells are positive and the others are negative because they followed another fate. Could you please explain this in more detail? If you only picked regions with especially this marker, it would make sense, but this should be stated.

Best regards
